# Low-Cost Hyperspectral Imaging in Macroalgae Monitoring

**DOI:** 10.3390/s25092652

**Published:** 2025-04-22

**Authors:** Marc C. Allentoft-Larsen, Joaquim Santos, Mihailo Azhar, Henrik C. Pedersen, Michael L. Jakobsen, Paul M. Petersen, Christian Pedersen, Hans H. Jakobsen

**Affiliations:** 1Department of Ecoscience, Marine Diversity and Experimental Ecology, Faculty of Science and Technology, Aarhus University, 4000 Roskilde, Denmark; mihailo.azhar@ecos.au.dk (M.A.); hhja@ecos.au.dk (H.H.J.); 2Department of Electrical and Photonics Engineering (DTU Electro), Technical University of Denmark, 4000 Roskilde, Denmarkhcpe@dtu.dk (H.C.P.); mlja@dtu.dk (M.L.J.); pape@dtu.dk (P.M.P.); chrp@dtu.dk (C.P.)

**Keywords:** hyperspectral imaging, artificial intelligence, macroalgae, spectral analysis, classification, biodiversity, remote sensing, 1D convolutional neural network

## Abstract

This study presents an approach to macroalgae monitoring using a cost-effective hyperspectral imaging (HSI) system and artificial intelligence (AI). Kelp beds are vital habitats and support nutrient cycling, making ongoing monitoring crucial amid environmental changes. HSI emerges as a powerful tool in this context, due to its ability to detect pigment-characteristic fingerprints that are often missed altogether by standard RGB cameras. Still, the high costs of these systems are a barrier to large-scale deployment for in situ monitoring. Here, we showcase the development of a cost-effective HSI setup that combines a GoPro camera with a continuous linear variable spectral bandpass filter. We empirically validate the operational capabilities through the analysis of two brown macroalgae, *Fucus serratus* and *Fucus versiculosus*, and two red macroalgae, *Ceramium* sp. and *Vertebrata byssoides*, in a controlled aquatic environment. Our HSI system successfully captured spectral information from the target species, which exhibit considerable similarity in morphology and spectral profile, making them difficult to differentiate using traditional RGB imaging. Using a one-dimensional convolutional neural network, we reached a high average classification precision, recall, and F1-score of 99.9%, 89.5%, and 94.4%, respectively, demonstrating the effectiveness of our custom low-cost HSI setup. This work paves the way to achieving large-scale and automated ecological monitoring.

## 1. Introduction

Kelp beds and macroalgal communities are often acknowledged for their ecological significance, as they play a crucial role by providing various biological services [1]. They function as essential habitats to different animal groups, including fish, and are widely used as shelter by organisms such as fish and crustaceans in their vulnerable juvenile life stages [2]. Moreover, macroalgal communities, especially brown algae forests, contribute to nutrient cycling and are among the most productive ecosystems on Earth as regards primary production and carbon sequestration [1,3]. Climate change-induced ocean warming and the general decrease of oceanic pH are leading to a decline in global macroalgal populations [1,4,5]. As a consequence, these losses threaten to destabilize marine ecosystems and, therefore, to profoundly disrupt biodiversity as well as human societies reliant on marine ecosystem services, particularly fisheries [5,6].

Continuous monitoring of macroalgal communities through remote sensing and imaging has become an important tool for understanding and documenting ecological changes in these habitats [7,8]. The synergistic integration of Artificial Intelligence (AI) with marine surveying and image processing techniques has demonstrated promising results in macroalgal monitoring and may enable the automation of taxonomic identification tasks, which in turn unlocks larger scale and more accurate monitoring surveys [9,10]. AI-driven species recognition has been commonly built upon standard RGB (Red, Green, Blue) images, due to the widespread availability of inexpensive commercial imaging systems with high data throughput. Despite the availability of immensely large datasets (required for model training) propelled by such systems, AI-driven predictions of marine environments remain limited to low-specificity taxonomic levels, due to the tremendous complexity of these environments, diverse species morphology, and chromatic similarities [11,12].

In recent years, Hyperspectral Imaging (HSI) has been proposed for marine monitoring, as an alternative to RGB cameras, to perform more accurate and detailed taxonomic investigations [12]. Hyperspectral cameras create three-dimensional data structures called hypercubes (or datacubes) by combining two-dimensional spatial information (x,y) with spectral information (λ). The two-dimensional data is stacked with each slice representing a monochromatic image at a well-defined wavelength channel [13]. This ability to capture high-resolution spectral information at the pixel level makes HSI suitable for taxonomic classification. While RGB systems acquire information in three broadband channels, HSI systems sample tens of bands, thus providing comparatively finer data from the scenes and more informative inputs to classification models [14]. Due to the higher level of spectral detail contained in the hypercubes, a smaller amount of labeled samples are needed for the AI processing [11,12]. This has proven useful in marine environments where samples are often less accessible and more expensive to obtain compared to land-based sampling [15].

Hyperspectral imaging is advantageous in a variety of fields, as it can reveal subtle biochemical and physiological changes that are both invisible to the naked eye and imperceptible to standard RGB systems. This optical technology has proven accurate in benthic habitat mapping and taxonomic classification of marine organisms, where color and morphological features obtained with standard cameras are often too coarse to capture the complexity of the organisms [16,17,18]. In the context of kelp bed monitoring, HSI can prove a powerful tool due to its ability to identify species-specific pigmentation that differs among distinct macroalgae, such as photosynthetic pigments (e.g., chlorophyll). The unique, but subtle pigment composition of each species can partly be captured by HSI, making classification possible even when different groups have overlapping spectral features [16,19]. Such knowledge has the potential to expand our understanding of marine ecosystems through detailed spectral analysis [12,20] and subsequently steer preservation efforts towards effective management of these ecosystems. Notwithstanding, practical large-scale deployment of HSI systems in field settings is currently hindered by the high costs and limited customization options of commercial cameras, particularly when it comes to spectral range and spectral-spatial resolution [21,22]. To address these cost barriers, alternative HSI solutions such as the use of a linear variable spectral bandpass filter (LVSBPF) have been explored [23,24,25,26]. These systems, while offering a simplified optical design and cost reduction, have not yet incorporated field-friendly consumer cameras like the GoPro, nor have they been optimized for marine applications. These challenges emphasize the need for technological advancements and cost-effective solutions to unlock the full potential of HSI in the aforementioned context.

In this work, we introduce a cost-effective HSI design that combines a readily accessible GoPro camera with an LVSBPF. We present and characterize the optical system and describe the pre-processing workflow required to convert a sequence of raw images into a hypercube. Finally, we demonstrate the capabilities of the camera in the examination and classification of different macroalgae species with overlapping spectral features, by feeding the spectral data into a previously-trained 1D Convolutional Neural Network (CNN).

The remainder of this paper is structured as follows: Section 2 details the optical setup and acquisition procedure and describes the data pre-processing pipeline used to construct hyperspectral cubes. Section 3 presents the spectral library and classification results obtained using the 1D-CNN model. Finally, Section 4 discusses spectral and classification results, as well as the implications of our findings for scalable, low-cost ecological monitoring, and outlines potential directions for future work.

## 2. Materials and Methods

### 2.1. Hyperspectral Camera

#### 2.1.1. Setup and Camera Settings

Figure 1 shows a schematic diagram of the developed HSI system. The core component of the hyperspectral system is an LVSBPF (for details about the filter, please find the 400–700 nm, 3–9 nm bandwidth model at Delta Optical Thin Film A/S, Copenhagen, Denmark) that acts as a dispersive element and, therefore, as the wavelength-selective element. This filter exhibits a spatially-dependent bandpass response, with a length of 47.8 mm and a linear gradient of center wavelength (CWL, $\lambda_{0}$) from around 400 to 700 nm (compatible with the optical transmission window of water [27]), and a nominal full-width at half-maximum (FWHM) spectral bandwidth increasing from 3 to 9 nm, respectively. The LVSBPF unlocks a cost-effective solution, by drastically reducing costs when compared with currently available commercial hyperspectral cameras. Furthermore, it enables fast frame rates by linearly sweeping the filter in front of a high frame rate consumer camera, therefore also enabling spatially coherent imaging, as the field-of-view (FOV) remains static throughout the acquisition of a full 3D datacube, which in turn facilitates the post-processing, since complex geometrical corrections and stitching are not required, as is the case with push-broom systems [28,29].

To obtain a stack of hyperspectral images (hypercube/datacube), the filter is mounted on a linear stage with a stepper motor (uStepper S32) with a 15-bit encoder and periodically translated in front of an off-the-shelf consumer camera. To evaluate the stability of the stepper motor mechanism with the LVSBPF, repeated measurements of a fixed spatial point across three full spectral sweeps were compared, showing minimal variation in the resulting spectra, thus indicating reliable filter performance under the defined scanning parameters. We opted for a GoPro Hero 11, due to its compact size, affordability, high-resolution, fast video modes, integrated functionalities, and ease of operation. For maximum spectral and spatial resolution in the hyperspectral imaging system, the filter must be mounted in an intermediate image of the object, and the total angular range of rays incident onto the filter must be limited to within ±5 deg. For this reason, we constructed an optical system around the filter, to be mounted in front of the built-in lens of the GoPro camera. The optical system implemented an intermediate image plane at the filter that at the same time eliminated the contribution from the field-of-view angle to the angular range of incident rays. In this case, the angular range of rays incident onto the filter was determined entirely by the numerical aperture of the lens system. The optical system comprises an input circular aperture (A), with a clearance diameter of a maximum of 8.7 mm (to ensure an angular incidence of a maximum of 5° at the filter), as well as a pair of 01” lenses (L_3_ with f = 50 mm, Edmund Optics 32-323, York, United Kingdom L_2_ with f = 25 mm, Edmund Optics 65-553, York, United Kingdom) that collimate the input light to be afterward transmitted through the LVSBPF located at the imaging plane. The light is afterward re-imaged by L_1_ (Edmund Optics 65-553) in combination with the GoPro-built-in lens, having an effective focal length of 2.69 mm. Due to the broadband nature of the hyperspectral system, we employed achromatic doublet lenses and corrected for spherical and chromatic aberrations.

#### 2.1.2. Operation and Data Acquisition

The GoPro camera is operated in continuous video mode and directly controlled using the Quik app. A sequence of static images of the scene is acquired and directly stored on a microSD card that can be accessed a posteriori for processing, e.g., with a computer. The LVSBPF is translated to scan the full spectral range across the full FOV. At each instant, i.e., for each frame, each row of the camera observes a distinct wavelength channel, corresponding to a specific lateral position on the LVSBPF surface, and translation of the filter accomplishes the acquisition of a full data cube, with wavelength being multiplexed in time. Due to the nature of the scanning, the total number of spectral data points, i.e., the number of wavelength channels, is set by the ratio between the frame rate of the camera and the scanning speed of the LVSBPF. Hence, by changing the translation speed of the filter, the spectral resolution can be tuned according to need.

For the experiments with macroalgae demonstrated herein, the HSI system was mounted vertically and pointing downwards. The biosamples were immersed in a water container placed underneath the sensor at a fixed vertical distance of around 0.7 m (Figure 1) to ensure cross-consistency between different acquisition sequences, i.e., different samples. The GoPro camera was operated in Manual mode, recording 2704 × 1520 pixels images with fixed parameters: a 240 Hz frame rate, a 1/240 s exposure time, and an ISO of 100. The parameters where locked to ensure consistency across spectral measurements. The LVSBPF translation speed was such that spectral channels ranging from 70 to 350, depending on scanning speed, were sampled, thus representing a full datacube acquisition time of 240/70 ≃ 0.3 s to 240/350 ≃ 0.69 s. To enhance the signal-to-noise ratio (SNR) and achieve uniform illumination, two light-emitting diode (LED) matrices were mounted in the opposite lateral walls of the water container, and the measurements were performed in a dark room environment.

#### 2.1.3. Hyperspectral Camera Characterization

We characterized the HSI sensor to evaluate its performance and to perform corrections of the acquired images during pre-processing. In the first instance, we empirically measured the gamma function of the GoPro to afterwards perform a gamma correction of the output images. Ideally, the imaging sensor responds linearly to the number of incident photons at the pixel level, meaning that the digital intensity count in the resulting image is linearly proportional to the incoming light level. Nonetheless, in reality, this response is often non-linear, and it must therefore be characterized to artificially linearize the obtained images and make all pixels directly comparable for subsequent hyperspectral analysis. For this purpose, we projected 3 laser pointers with nominal center wavelengths of 405 nm (blue channel, Thorlabs PL256), 520 nm (green channel, Thorlabs PL201), and 635 nm (red channel, Thorlabs PL202) onto white cardboard and imaged the diffusely scattered light with the GoPro (no LVSBPF). A wire-grid linear polarizer (Thorlabs WP25M-VIS) was used as a variable attenuator to control the laser power onto the target and, therefore, the photon budget in the camera plane. The recorded intensity in each corresponding RGB channel was analyzed for each power level and wavelength for the same camera settings as aforementioned (a 1/240 s exposure time, ISO = 100, 8-bit images). The resulting gamma curves are plotted in Figure 2. As observable, the response of the imaging sensor becomes highly non-linear as the digital counters approach the saturation limit (255 counts). This signifies that the relative pixel intensities must be corrected during the processing of the hyperspectral images to linearize their responses.

A similar procedure was employed to characterize the spectral resolution of the system. The fundamental spectral resolution is, ultimately, defined by the design of the LVSBPF and the respective FWHM. However, in practice, the spectral response is broadened due to the lateral displacement of the filter during the time window corresponding to one exposure time, as well as the magnification of the optical system (i.e., the projected pixel size on the filter plane). For this estimation, the 520 and 635 nm monochromatic laser pointers were aligned perpendicularly to the translation direction of the filter to be observed in a common wavelength channel at any time. The 405 nm laser was not utilized since it induced broadband autofluorescence on the white cardboard. The spectra of the lasers were measured a priori using a calibrated spectrometer (Avantes, AvaSpec-2048-FT-SPU, Colorado, USA, 1.2 nm spectral resolution). Afterwards, video footage was acquired at 240 fps (ISO = 100), while the LVSBPF was scanned in the GoPro’s FOV.

After acquisition, the footage was processed in Matlab, R2024b. In the first stage, the RGB channels were integrated at the pixel level to generate a single-channel image. Posteriorly, the temporal intensity profile of each projected laser spot was analyzed. Since the LVSBPF is moving continuously, each frame corresponds to a specific wavelength, and the obtained response is a convolution of the laser spectrum and the moving filter transmission properties. The intensity peaks when the LVSBPF’s CWL, $\lambda_{0}$, at the spatial location of the laser spots matches the peak wavelength of the laser, and then quickly decreases as the filter is spatially convoluted. The frame number (i.e., time stamp) was mapped into the wavelength by using the known laser spectra and assuming a linear relationship (a constant speed of the linear translation stage + the linear wavelength sweep of the LVSBPF). The results are plotted in Figure 3. The computed spectral resolution of the hyperspectral camera is Δλ = 7.2 nm at $\lambda_{0}$ = 514 nm and Δλ = 9.4 nm at $\lambda_{0}$ = 632 nm. In practice, the lasers are not monochromatic light sources, so the resolution is overestimated, as it represents a convolution of the laser’s bandwidth with the actual response of the HSI system.

### 2.2. Study Area and Sample Collection

The samples analyzed in this work were taken from the beach in Hornbaek Plantage (56 05′31″ N, 12 29′10″ E), shown in Figure 4, due to the easy accessibility and relatively high biodiversity of macroalgae. The beach is rocky with sporadic bigger rocks in the tidal zone. The bottom changes from rocky to sandy after a few meters, with patches of kelp in the tidal zone and scattered macroalgae patches in deeper waters (>2 m). The water varies in turbidity, and the salinity ranges from 1.39% to 1.62%.

Sampling was conducted twice in separate months to capture seasonal changes (April and July 2024), and close to the shore at a depth from 0.2 to 2 m, where macroalgae were more numerous. The samples were collected by hand into a mesh bag using snorkeling equipment (wetsuit, gloves, hood, fins, weight belt, mask, and snorkel). Two representatives of brown macroalgae and red macroalgae with overlapping spectral features were selected (Figure 5). The sampled set included 11 specimens of *Fucus serratus*, 12 specimens of *Fucus versiculosus* (both brown macroalgae), 9 specimens of *Ceramiun* sp., and 11 specimens of *Vertebrata byssoides* (both red macroalgae). *Vertebrata byssoides*was only sampled in July. The macroalgae were sorted and stored in plastic containers with saltwater from the site. The samples were then transferred to the laboratory at the Technical University of Denmark (DTU), Risø, 399 Frederiksborgvej, 4000 Roskilde. When the samples were not directly used for HSI, they were stored in aquariums filled with saltwater and with air diffusers, in a climatic room at 5–10 °C with LED lighting, to keep the samples fresh for later experiments.

#### Taxonomic Identification

The species that could not be identified on site were examined a posteriori in the lab. The samples were placed in freshwater and identified to the lowest taxonomic level using a Leica Wild M3C stereo microscope, Wetzlar, Germany, with an external light source and a Wild Heerbrugg Plan 1.0× objective, and two catalogues: *Havets Dyr og Planter* [30] and *Danmarks Havalger*–*Rødalger* (*Rhodophyta*) [31]. Afterwards, the samples were stored in a plastic container with a solution of 70% ethanol and 30% salt water, labeled with their respective ID and sample information.

### 2.3. Pre-Processing of Hyperspectral Data

The videos acquired for each sample consist of a sequence of images (or frames) acquired at 240 Hz. The raw data consists of a four-dimensional stack, with two spatial coordinates (x,y), three color channels (RGB), and one temporal dimension (frame number). All pre-processing was performed in Python (3.11.6).

In the first stage, we selected the frames corresponding to a full scan of the LVSBPF through the FOV, and, therefore, a full single spectrum. Afterwards, the spatial dimensions (x,y) were cropped to a 469 × 392 or 726 × 544 image, depending on the size of the raw image, to select the Region of Interest (ROI) corresponding to the underwater scene. A simplified diagram of the workflow is seen in Figure 6. After cropping and loading the stack of images, the stack of images was linearized using the gamma functions previously obtained (Figure 2).

Due to the gradient of the filter, on each frame, each row of the image corresponds to a distinct wavelength channel and each frame displays a color gradient from top to bottom (along the (***y***) direction, Figure 7A). This means that there is a shift in the spectra according to the analyzed row of the image. Since the filter has a linear gradient and is moving at a constant speed, this shift is linear. Therefore, this shift needs to be corrected to re-order the stack of images and create a new stack of fully monochromatic images and, henceforth, to obtain a 3D hypercube with dimensions (x,y,λ) (2 spatial dimensions, and wavelength). To correct the filter gradient, we used a grey-colored plate, i.e., with a flat reflectance spectrum (color neutral), as a background for all the acquisitions. We then selected two pixels on the same *x* coordinate (i.e., the same column), but vertically displaced (i.e., different rows): one at the top of the plate and another at the bottom. Subsequently, we cross-correlated the intensity versus the frame number spectrum of the two pixels for each one of the RGB channels to compute the correlation peak that quantifies the shift in a certain number of frames (i.e., the number of frames in which the bottom pixel observes the same wavelength as the top pixel). Finally, using linear regression, we obtained the linear function that determines the shuffling of rows between frames to create fully monochromatic images and eliminate the wavelength gradient intrinsic to the filter (Figure 7B).

The GoPro camera acquired RGB images, meaning that each pixel quantifies the intensity in 3 distinct channels. Each color channel was trimmed to remove baseline noise from the spectrum (red: 510 to 660 nm, green: 465 to 620 nm, and blue: 430 to 566 nm). Nevertheless, the LVSBPF acted as the color-selective element in our HSI camera, so the intensity of the trimmed RGB channels was integrated (i.e., summed) to provide a single and total pixel intensity, without a loss of information, and thus a single-channel spectrum.

The next step in processing involved converting frames to corresponding wavelengths. We began by measuring the predefined red and blue offset locations in the RGB spectrum on the grey-colored plate, which corresponded to λ=642 nm ($\lambda_{642}$) at the end of the red spectrum and λ=436 nm ($\lambda_{436}$) at the start of the blue spectrum.

For each sample, we determined the onset values by identifying the frame numbers at the start of the blue spectrum and the end of the red spectrum on the grey-colored plate. These onset values corresponded to the respective offset values. To convert frames to wavelengths, we used a linear function, calculated as follows:(1)$$m_{\lambda }=\frac{\lambda_{642}-\lambda_{436}}{{\mathrm{frame}}_{{\mathrm{red}}_{\mathrm{onset}}}-{\mathrm{frame}}_{{\mathrm{blue}}_{\mathrm{onset}}}}$$(2)$$b_{\lambda }=\lambda_{436}-m_{\lambda }\cdot {\mathrm{frame}}_{{\mathrm{blue}}_{\mathrm{onset}}}$$
where $m_{\lambda }$ is the slope, $b_{\lambda }$ the y-intercept, and $\lambda_{642}$ and $\lambda_{436}$ are the wavelengths for the red and blue offsets, respectively.(3)$$\lambda =m_{\lambda }\cdot \left(\mathrm{frames}\right)+b_{\lambda }$$
where frames represent the frames obtained by the scan.

After the wavelength conversion, each frame corresponded to a single wavelength channel. Since we wanted the full spectrum, we interpolated the resulting data using linear interpolation to obtain intensity values at each wavelength in the 400–700 nm range. We discarded wavelength bands below 435 nm and above 699 nm due to a low SNR. We then performed a reflectance calibration of the data to bring it to a universal scale in which hypercubes from different samples can be directly compared (externally consistent), since reflectance is an intrinsic property of the objects and, thus, it is independent of the spectrum of the illumination source, the transmission properties of the LVSBPF, the properties of the water column, and the image parameters (e.g., exposure time) [32]. For calibration, we used the color-neutral grey background plate, with a flat reflectance spectrum (R = 0.1), as described in [32]. After calibration, a Savitzky–Golay filter with an order of 2 and a window length of 9 was applied to denoise the spectra of the hypercube [33]. Finally, we binned the wavelength channels by a factor of 5, to reduce noise without loss of spectral information [34] (as the wavelength resolution/step before binning was well below the FWHM of the system) and to reduce the data size. These processing steps are shown in Figure 7C for a sampled point of a brown macroalgae. It is also noteworthy to mention that, due to this processing workflow, precision is not mandatory when cropping the initial video to select the frames corresponding to a single scan of the filter, as long as one ensures that a full cycle is selected.

### 2.4. Classification Model

#### 2.4.1. Software and Computer Specifications

All models were made using Python (3.11.6). Experiments were made on a computer with a CPU with the following specifications: 13th Gen Intel(R) Core(TM) i9-13900K 3.00 GHz, RAM: 64.0 GB, GPU: Nvidia GeForce RTX 4090.

#### 2.4.2. Data Description

The taxonomic identification of macroalgae is a supervised machine-learning problem, as one desires to build a classification model that can output a class from an input spectrum. This requires expert-annotated data for both training and testing the model. The test data set includes four images, one for each macroalgae class, that remain unseen by the model during training and hyperparameter optimization. To annotate the hyperspectral images, we used bounding boxes whose pixel locations were determined from the corresponding RGB images. The software *Roboflow, 2024* [35] was used for this step. Bounding boxes were drawn to include only macroalgae while avoiding areas overgrown with epiphytes and boundaries with the background. The background was also annotated. The dataset distributions are described in Table 1.

#### 2.4.3. Model Description and Training Parameters

Based on previous studies on the classification of hyperspectral data [36,37,38], a one-Dimensional Convolutional Neural Network (1D-CNN) was modeled using the PyTorch library (v2.1.2). A CNN model comprises unique convolutional layers used to automatically extract features via filters. The number and size of these filters are expressed as the kernel number and kernel size, respectively. Among these layers are pooling layers, which reduce the size of the feature maps (data compression), consequently making the neural network less sensitive to small changes in the data. The pooling layers are defined by size and stride. The most common type is Max pooling, where a small window defined by size (e.g., 2 × 2) slides over the feature map and retains only the maximum value within each window. The stride size defines how far the window moves. To help the model learn complex patterns in the data, an activation function (most commonly ReLu) introduces non-linearity by changing negative values in the feature map to zero. After the convolutional and pooling layers, the output is flattened via a flatten layer and fed to dense layers that connect the features to neurons that use these connections to classify images. The architecture of our CNN largely follows that of [38]; however, the third convolutional and pooling layers were removed, as initial experiments showed degraded performance with this additional layer. Our model takes in data in the form of a 1D array of reflectances (with wavelength being encoded in the position in the array) and consists of one input layer, two convolutional layers, two pooling layers, one flatten layer, and two dense layers. In the case shown herein, the input vector length is 45. A schematic diagram is presented in Figure 8.

The first convolutional kernel size was 3, the stride was 1, and the number of kernels was 32. The second kernel size was 3, the stride was 1, and the number of kernels was 64. For each convolutional layer, the padding was set to 1. ReLu was set as the activation function for this model. After each convolutional layer, we reduced the number of parameters by using a max pooling layer. The max pooling layer size was 2, and the stride was 2. A dropout was set after each pooling layer with a p-size of 0.25. The output of the pooling layer was then flattened into a 1-D array through the flatten layer, and connected to two dense layers with 128 neurons in each. A linear function was used in the output layer to produce a single class from the four possible species of macroalgae and background. We used a cross-entropy loss function in the 1D-CNN, which is described in [38].

The dataset was split into 90% for training and 10% for validation to support hyperparameter tuning. Alternative splits of 80/20 and 70/30 (training/validation) were also tested, but these configurations led to reduced model performance, particularly in recall and F1-score. To identify the optimal number of epochs for training our CNN model, we monitored the model’s performance on both training and validation datasets over successive epochs using their loss curves to avoid overfitting. After each epoch, we recorded the training loss and validation loss. We examined the plots and noted the number of epochs where the curves stabilized. This number of epochs was then used to train the final CNN model. The final model was trained with 300 epochs and an Adam optimizer with a batch size of 256. The data set was well-balanced, so all classes were weighted equally.

#### 2.4.4. Model Evaluation

We assessed the model’s performance by presenting it with our test data set (individual macroalgae not used in training). The precision, recall, and F1-score metrics (calculation formulas are described in [38]) were quantified for each class by comparing the labeled ground truth data (explained in Section 2.4.2) with the prediction of the model. We took the average of each metric (precision, recall, and F-1) across the four test-macroalgae images to evaluate the model as a whole.

## 3. Results

### 3.1. Spectral Library

The mean reflectance spectra of the annotated macroalgae pixels used to train the model are shown in Figure 9. Overall, the spectra disclose optical fingerprints that are distinct among the four classes studied in this work. The spectra show two local maxima absorption peaks (and therefore local minima in reflectance) for both brown macroalgae at 445 nm and 641 ± 3 nm and two lesser peaks at 496 and 603 nm. The two red macroalgae had two local maxima absorption peaks at 440 and 639 nm. The red macroalgae *Ceramium* sp. had two lesser absorption peaks at 491 and 603 nm, whereas the red macroalgae *V. byssoides* showed peaks at 481 and 598 nm. The brown macroalgae spectra exhibit similar structures and intensity levels with the exception of a small intensity variation at the red part of the spectra with *F. serratus* having higher intensity. The two red macroalgae exhibit similar spectral structures with a slightly shifted tendency. *Ceramium* sp. had a generally higher absolute reflectance than *V. byssoides*. Nonetheless, the two macroalgae have higher reflectance in the red part of the spectrum, which is consistent with their color. Generally, the red macroalgae species were different in their spectral structure and had higher reflectance compared to the brown macroalgae. All of the macroalgae, however, have relatively low reflectance, resulting in a dark appearance. Each spectrum revealed a global reflectance peak at the longer wavelengths.

### 3.2. Model Performance and Classification

The 1D CNN model was trained on the training set (see Table 1) for 300 epochs, based on learning curve analysis, with a batch size of 256. The total training time was approximately 450 min. Increasing the batch size exponentially from 32 to 256 resulted in faster training times and improved the overall performance metrics. Training beyond 300 epochs led to overfitting. Furthermore, pixel-level reflectance calibration (see Section 2.3) substantially impacted the model’s performance. The model was tested on four HSI images excluded from the training set, each representing one of the target macroalgae classes. The classification results are summarized in Table 2. The model demonstrated strong performance, with precision, recall, and F1 scores exceeding 0.9000 for all classes, except for the recall of *V. byssoides*(0.8895) and *F. versiculosus* (0.8794). The average metric scores across all classes were as follows: Precision—0.9997; Recall—0.8955; F1 Score—0.9447. Given our small dataset, the results are promising in that HSI can discern between visually similar macroalgae species.

A visual analysis of the segmented predictions shown in Figure 10 revealed that the model successfully classified a majority of the pixels corresponding to macroalgae within each test image. The model achieved the highest segmentation accuracy for the two brown macroalgae, while red macroalgae exhibited fewer correctly segmented pixels. In particular, predictions for the red macroalgae *V. byssoides* showed small areas segmented between the gaps of the brown macroalgae *F. versiculosus*. Overall, the majority of pixels in each test image were correctly classified, with some degree of overlap between classes, as illustrated in Figure 10.

## 4. Discussion

This study has shown that, by using an inexpensive and custom hyperspectral camera setup that combines a consumer-grade GoPro camera with a scanning linearly variable spectral filter, we were able to classify morphological and spectrally overlapping macroalgae species using a 1D-CNN model.

### 4.1. Spectral Library

The spectral library comprises spectra from two brown and two red macroalgae. The species within each group share similar features, namely characteristic pigmentation and morphology, and were chosen because of the difficulty in identifying them on site. This study showed that, by using a GoPro Hero 11 camera with the scanning LVSBPF, we were able to obtain a reflectance spectrum of different macroalgae. As expected, the spectra within each macroalgae group showed similar structures with the brown macroalgae showing almost identical spectra likely due to their close taxonomic relation. The examined brown macroalgae share the same ecological niche in the intertidal zone and are often found growing side by side. This means that biotic and abiotic factors that can cause variation in their pigmentation ratio and concentration are likely the same. However, studies show a seasonal variation of the pigments and that they vary differently between *F. serratus* and *F. versiculosus* [39]. This also means that, when conducting studies like this, one has to integrate a seasonal variation in their data, since the pigmentation concentration affects the macroalgae spectrum, and thus the input data analysis for the classification model. The brown and red macroalgae spectra all showed an absorption peak at around 445 and 640 nm, corresponding to that of chlorophyll-a (a light-harvesting pigment primarily involved in photosynthesis) [40]; however, our spectra showed a lower absorption peak at the red light than is normally registered for chl-a (665 ± 15 nm) [32], which is likely an artefact from the scanning filter. The pigment fucoxanthin, a dominant carotenoid associated with brown macroalgae and responsible for light-harvesting and protection against UV, has a peak at around 450 and a lesser peak at 480 ± 10 [40,41]. This corresponds to our results where brown macroalgae showed a higher absorption peak at the beginning of the spectra compared to red macroalgae because both chl-a and fucoxanthin excitate the most in this spectral area. Both red macroalgae exhibited a local absorption peak at around 600 nm, which has been linked to a satellite band of chl-a in red macroalgae [42]. This study has shown that, with limited cost, one can perform HSI to create spectral libraries. Such libraries contribute to our understanding of the role of macroalgae in their ecosystems and how they respond to environmental changes. Depth, salinity, turbidity, and temperature are all abiotic factors that impact the macroalgae distribution and influence the composition of their pigments, which will be visible in their spectrum as well [39,43].

### 4.2. Model Performance and Classification

In our study, we employed a 1D-CNN model with two convolutional layers. This gave the best results compared to three convolutional layers as was mentioned in [38]. The third layer resulted in poor prediction accuracy, potentially due to overfitting on the current dataset. However, with larger or higher resolution 16-bit images, a third layer is likely to enhance model performance. Each class exhibited near-perfect precision (>0.998) with slightly reduced recall (∼0.879–0.912). These results indicate that the model is highly adept at correctly identifying macroalgae samples, with minimal misclassification, particularly in terms of precision. However, the slight drop in recall for all classes suggests that the model occasionally fails to identify true positives, potentially due to subtle intra-class variations or challenges in distinguishing fine morphological differences among closely related macroalgae species. Notably, the model performed well in distinguishing the spectrally similar species *F. serratus* and *F. versiculosus*, though this pair exhibited the greatest degree of overlap in the results (Figure 10). The results prove the high discriminatory power of using HSI. However, the amount of data generated, given that each image or pixel is multiplied by the number of wavelengths captured, requires substantial computational resources. Model sensitivity to different pre-processing features, particularly pixel calibration, must be carefully managed since changes in the imaging setup (i.e., adjustment of the aquarium position or lighting) can affect the results. The study aimed to classify macroalgae that are challenging to distinguish even with the naked eye and thus we chose red filamentous macroalgae and closely related brown macroalgae. Since the model takes input solely from spectral data, avoiding epiphytes is crucial to minimize spectral contamination. While *Fucus* species allow for easy epiphyte removal, red filamentous macroalgae present a greater challenge, since their epiphytes grow intertwined with the macroalgae itself and are normally only detectable under a microscope. To mitigate this, we made sure to include as much cleaned algal material as possible to reduce the risk of epiphyte-derived spectral inference. Overall, the model demonstrates significant potential for accurate macroalgae classification, with room for improvement in enhancing recall, which could be addressed by further optimizing the model or including more diverse training data. These adjustments would help the model’s capacity to identify challenging species and address spectral overlap more effectively.

### 4.3. Implications and Perspectives

The cost of the camera setup included 399 EUR for the Gopro Hero (11), 187 EUR for the conveyor setup, 1200 EUR for the Delta spectral filter, and 415 EUR for the lenses. With a total cost of ∼2.200 EUR, we constructed an HSI camera that successfully classified four challenging macroalgae species based on their spectral features solely. The price of commercial HSI cameras starts at 28,000 US dollars, causing a barrier to in situ monitoring [44,45]. While commercial HSI cameras typically limit customization options, our approach allows for greater flexibility. It enables modifications of the setup, so that one can change the camera itself to meet specific requirements or even change the spectral filter if a different spectral range is needed. This is, to our knowledge, the first instance of using a consumer camera like GoPro for HSI. This demonstrates the potential for its application in ecological monitoring, providing a low-cost and customizable alternative to commercial HSI cameras. By offering an accessible, potentially automated solution powered by machine learning, this approach could significantly enhance the affordability of ecological monitoring, particularly for macroalgae studies.

Our system utilizes an RGB camera (GoPro Hero 11), which requires critical pre-processing steps to convert RGB data into HSI, a process that can seem complex. A way to simplify this is by using a monochromatic camera, as it bypasses the built-in color filter of the GoPro. This approach reduces pre-processing complexity by directly capturing the pixel intensity values without the need to separate the color channels (red, green, and blue) and allows for a higher bit depth. The GoPro, however, offers advantages such as 240 fps video, high resolution, a compact long-lasting battery, and remote control, making it ideal for field setups. Proper lighting is crucial in HSI systems to ensure pixel saturation and avoid spectral variations. We used two LED matrices powered by an external car battery to illuminate the aquarium. Our aim has been to explore the integration of a consumer camera in a low-cost HSI system using controlled experimental settings and isolating system performance by limiting external variables. Moving forward, adapting the system for underwater field use is ideal. Such a system will require a more compact design with lenses, a conveyor setup, a GoPro camera, and a filter in a waterproof container suitable for remotely operated vehicles (ROVs). Underwater deployment requires a much faster filter to ensure a full scan spectrum in a moving environment, especially when light variability and scattering present challenges. Reliable lighting and reference objects will help maintain pixel saturation in shifting water conditions. Integrating our HSI setup with ROVs will push the boundaries of automated large-scale ecology monitoring not only for macroalgae but also for other complex biological communities, contributing to future conservation efforts.

## Figures and Tables

**Figure 1 sensors-25-02652-f001:**
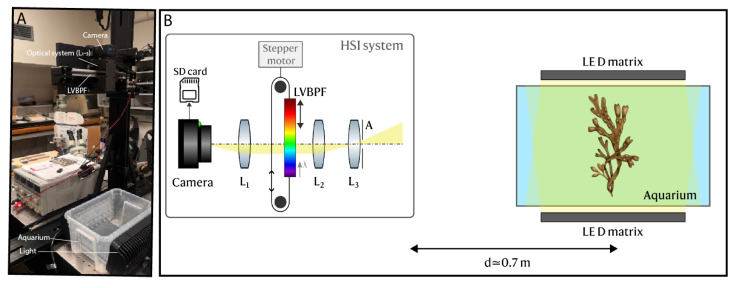
(**A**) Picture of the Hyperspectral Imaging (HSI) setup showing camera, optical system, continuous linear variable spectral bandpass filter (LVSBPF), aquarium with LED matrices. (**B**) Schematic diagram of the developed HSI system. A continuous linear variable spectral bandpass filter (LVSBPF) (red is 700 nm, blue is 400 nm) is mounted on a stepper motor linear stage and periodically translated in front of a consumer camera (GoPro Hero 11). An optical system (L_1_–L_3_) that implements an intermediate image plane at the filter and restricts the angle of incidence of the incoming light on the filter plane.

**Figure 2 sensors-25-02652-f002:**
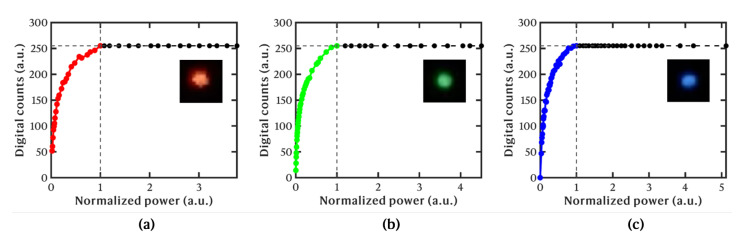
Gamma functions, i.e., digital intensity count versus normalized power (photon number), for each RGB channel of the GoPro camera: (**a**) red channel, measured with a 635 nm laser; (**b**) green channel, measured with a 520 nm laser; (**c**) blue channel, measured with a 405 nm laser. Max intensity at 255 Digital counts is indicated with a dashed point.

**Figure 3 sensors-25-02652-f003:**
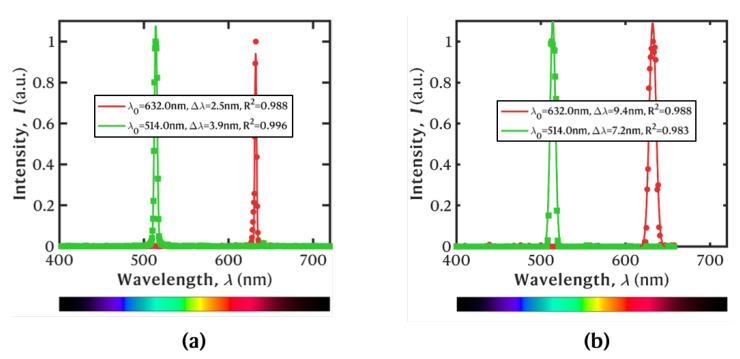
Spectrum of the laser pointers as measured (**a**) with a calibrated spectrometer and (**b**) with the Hyperspectral Imaging (HSI) camera.

**Figure 4 sensors-25-02652-f004:**
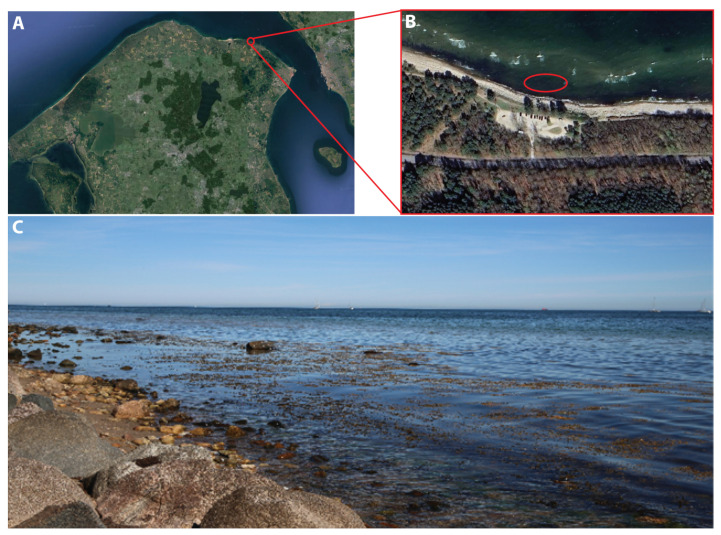
Sampling area. (**A**) Geoposition of Hornbaek, Northern Zealand, Denmark. (**B**) The sampling site at Hornbaek Plantage, 56 05′31″ N, 12 29′10″ E. The red circle indicates the sampling area. (**C**) Photo of the sampling site.

**Figure 5 sensors-25-02652-f005:**
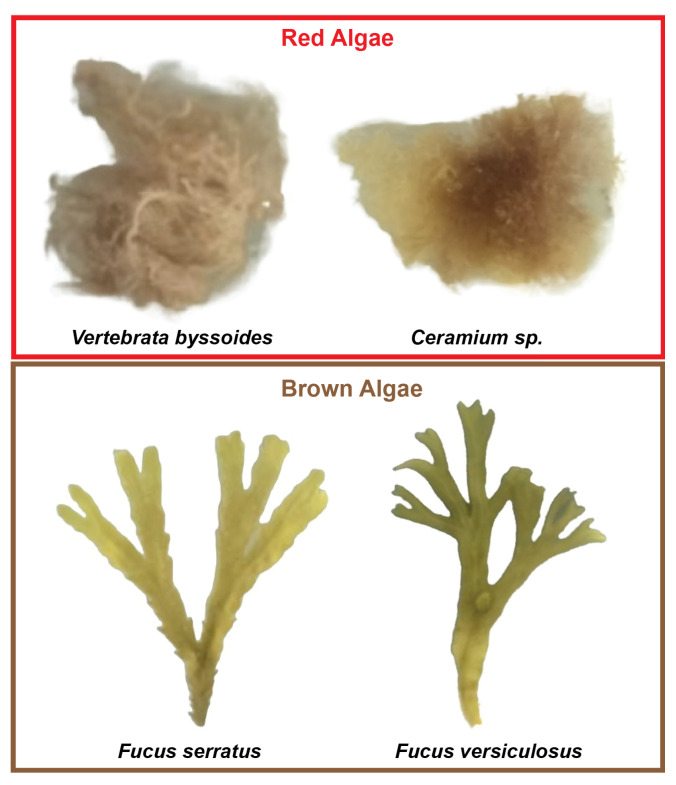
Macroalgae samples collected at Hornbæk Plantage during April and July 2024.

**Figure 6 sensors-25-02652-f006:**
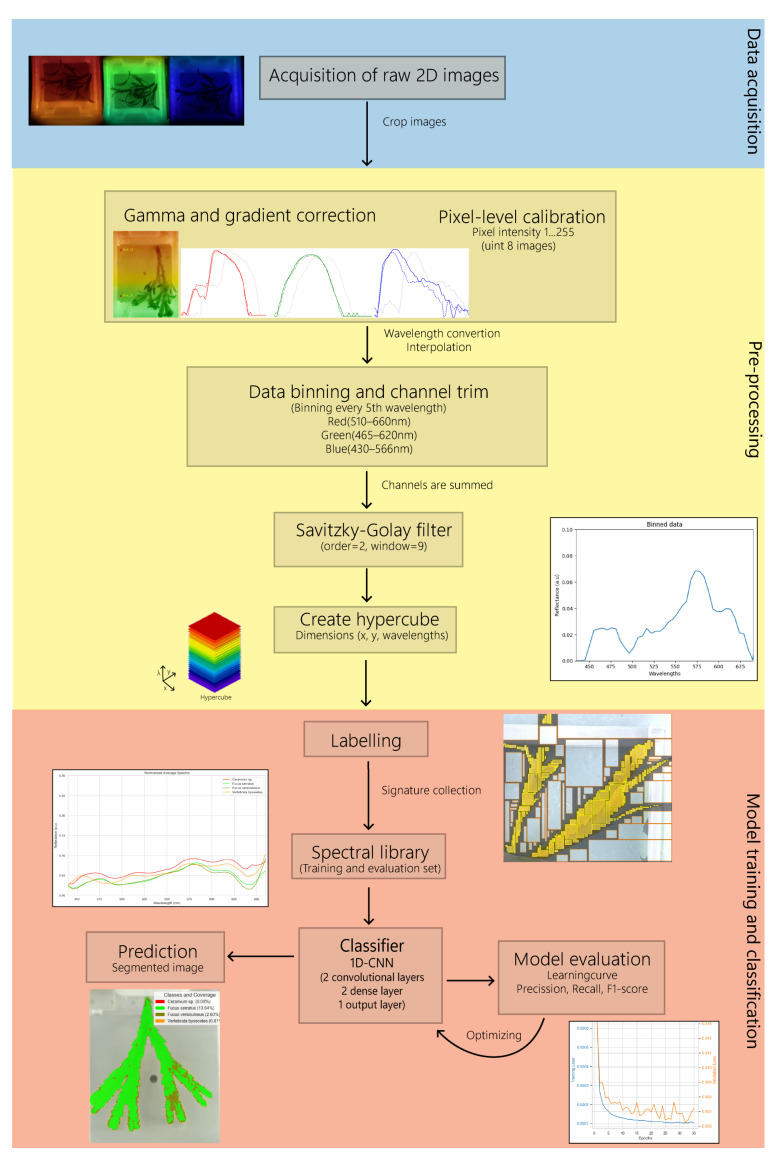
Work-flow chart. The chart shows the simplified workflow for 1. The acquisition of the images. 2. The pre-processing part, which includes gamma and gradient correction of the RGB channel to uniform the color across the image, data binning, Savitzky-Golay filter and creation of hypercube. 3. Labelling data to create spectral library and training and classification of algae using the 1D Convolutional Neural Network model (1D-CNN).

**Figure 7 sensors-25-02652-f007:**
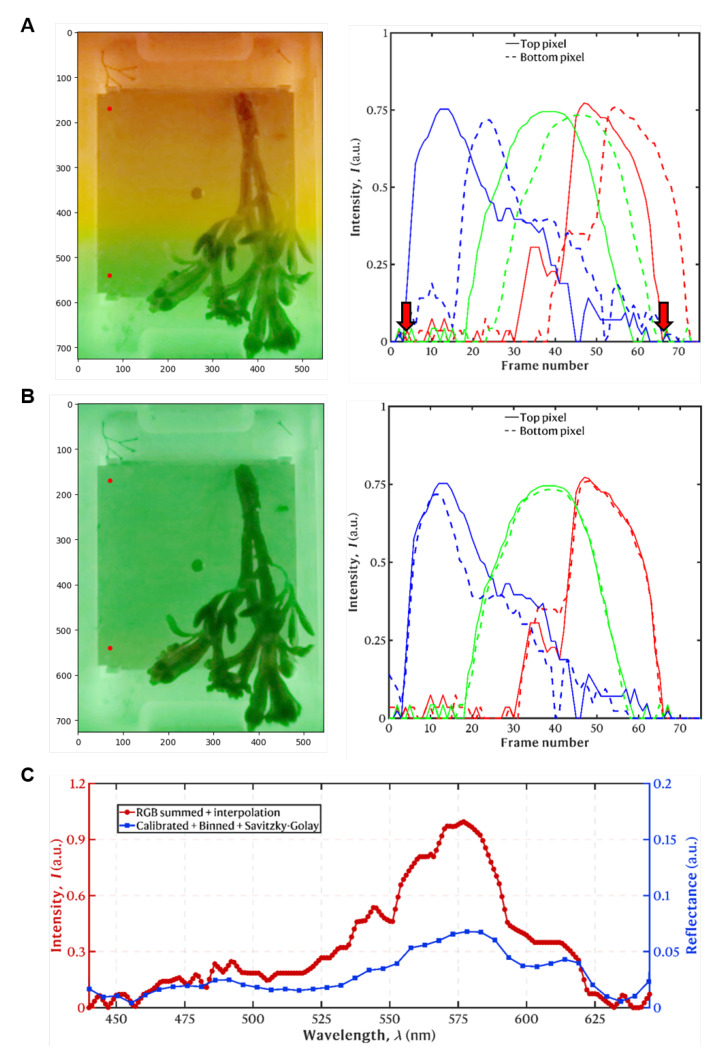
Pre-processing of spectral data. (**A**) Spectral gradient from the LVSBPF. (**Left**) Gamma-corrected image at frame 40 of brown macroalgae with two red dots showing the pixel positions at the top and bottom of the grey-colored plate. (**Right**) The graph shows the spectrum for each channel (RGB) for the grey-colored plate. The solid line indicates the top part of the panel, and the dashed line indicates the bottom of the panel, showing the gradient shift between the top and bottom. The red arrows indicate the blue and red positions for the onset and are taken at the start of the blue spectrum and the end of the red spectrum. The spectra are normalized. (**B**) Shifted spectra. The graph shows the shifted spectrum for each channel (RGB) of the grey-colored plate after cross-correlation. The solid and dashed lines are described above. The spectra are normalized. (**C**) The red plot shows the normalized summed spectrum of trimmed RGB channels with linear interpolation of wavelengths. The Y-axis shows intensity measured in artificial units (a.u.) shown in red. The blue plot shows the same spectrum after pixel calibration, data binning (factor of 5), and the Savitzky–Golay filter (order = 2, window = 9) is applied. The Y-axis shows intensity measured in arbitrary units (a.u.) shown in blue.

**Figure 8 sensors-25-02652-f008:**
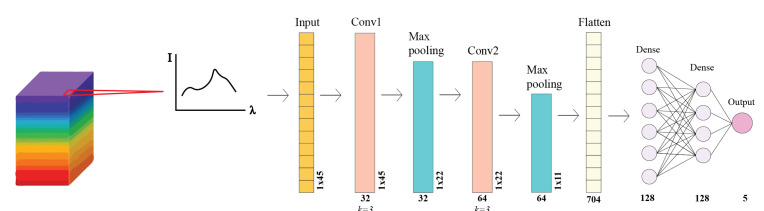
Architecture of the 1D-CNN trained for macroalgal classification. Our CNN takes in spectral information from 45 bands from each pixel in the hypercube (violet, 400 nm and red, 700 nm) in the training set in the form of a one-dimensional array (1 × 45). The model consists of two convolutional layers, two pooling layers, one flatten layer, and two dense layers. The name of each layer is shown at the top, kernel numbers are shown underneath the convolutional and pooling layers, and kernel size is shown as k. Output size is shown on the side of each layer and at the bottom for the flatten and dense layer and output.

**Figure 9 sensors-25-02652-f009:**
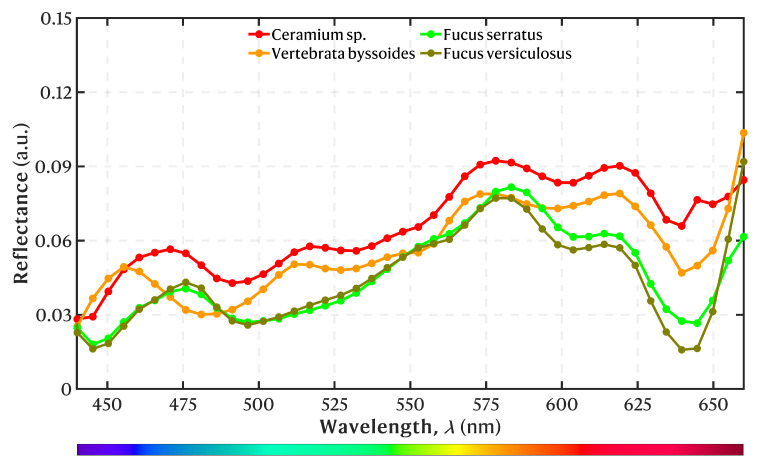
Mean reflectance spectra for each macroalgae class in the spectral library used as a training dataset for tuning the 1D-CNN model.

**Figure 10 sensors-25-02652-f010:**
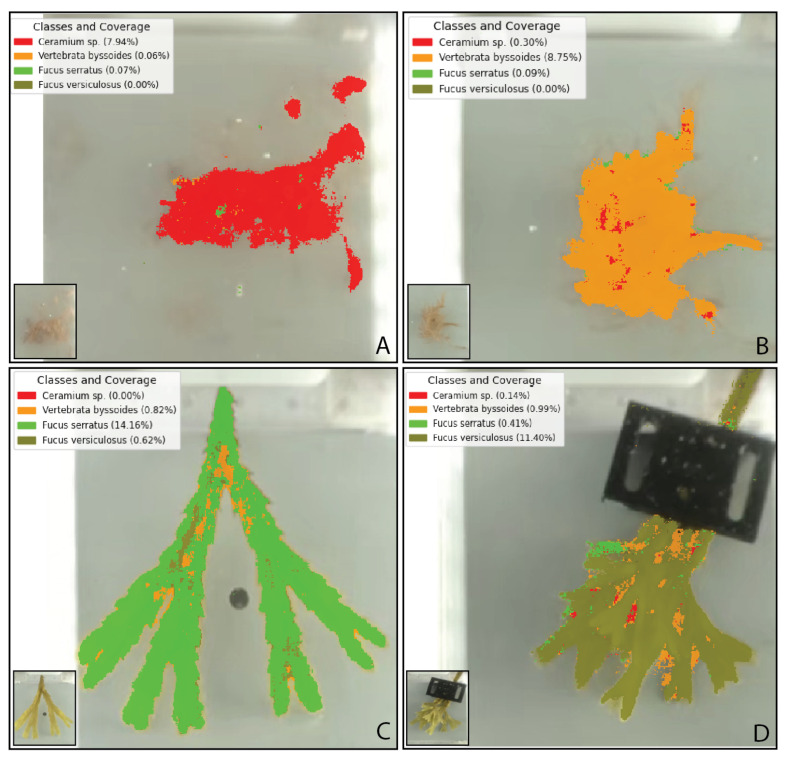
Classification results of each class generated by the 1D-CNN model. Predicted segmentation for (**A**) *Ceramium* sp., (**B**) *Vertebrata byssoides*, (**C**) *Fucus serratus*, and (**D**) *Fucus versiculosus*. The legend indicates the class and the corresponding coverage percentage predicted by the 1D-CNN model. For clarity, the background that represents the remaining coverage percentage is excluded.

**Table 1 sensors-25-02652-t001:** Distribution of the training and test datasets among the different classes (annotated pixels).

Color Class	Species	HS Images Train/Test	Training Datapoints	Testing Datapoints
Red macroalgae	*Ceramium* sp.	8/1	278, 103	9648
	*Vertebrata byssoides*	10/1	205,100	15,807
Brown macroalgae	*Fucus serratus*	10/1	269,536	16,555
	*Fucus versiculosus*	11/1	212,982	15,693
	**Total **	39/4	965,721	57,703

**Table 2 sensors-25-02652-t002:** Per class performance of the classification model on the test set.

Label (Species)	Precision	Recall	F1-Score
*Ceramium* sp.	1.0000	0.9010	0.9479
*Vertebrata byssoides*	1.0000	0.8895	0.9415
*Fucus serratus*	0.9999	0.9121	0.9540
*Fucus versiculosus*	0.9987	0.8794	0.9353
**Average score**	**0.9997**	**0.8955**	**0.9447**

## Data Availability

The data presented in this study are available on Figshare: https://doi.org/10.6084/m9.figshare.27619230, accessed on 20 February 2025.

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
