# Peer review of "Low-Cost Hyperspectral Imaging in Macroalgae Monitoring"

_sensors, 2025, doi:10.3390/s25092652_

Round 1
Reviewer 1 Report
Comments and Suggestions for Authors
The paper has the following issues
1. The paper proposes a low-cost hyperspectral imaging system that combines GoPro cameras with LVSBPF, which has significant cost advantages. And successfully classified brown algae and red algae with highly overlapping morphological and spectral features through 1D-CNN, verifying the feasibility of the technology and providing new ideas for large-scale ecological monitoring through hyperspectral methods. Suggest highlighting the improvements made to the model and experimental results in the abstract section to demonstrate its innovation and feasibility.
2. In terms of experimental design, the experiment was only conducted in a controlled water environment and the system performance was not validated in real ocean scenarios such as dynamic water flow, complex lighting, and suspended particle interference, which may overestimate the robustness of practical applications. And the dataset is divided into 90% for training and 10% for validating the trained model. What impact do different data partitioning methods have on the experimental results? Suggest the author to conduct appropriate analysis on the above two points.
3. From the experimental results, this article has provided a feasible low-cost solution for large-scale biological monitoring, and is innovative in the development and application of low-cost hyperspectral imaging technology. It is suggested to add a summary of the contributions made by each chapter of this article at the end of the introduction, in order to facilitate readers to have a general understanding of the system of this article during initial reading.
4. This article is a scientific and technological innovation paper. It is recommended that the author cite more papers from the past five years in terms of references to reflect the timeliness of this article. It is suggested that the author provide more detailed subheadings for each chapter of this article to reflect its innovative content.
5.The subheading is too simple, such as software, Model, Suggest the author to rename it to reflect the research content of each section.
Comments on the Quality of English Language
The English could be improved to more clearly express the research.
Author Response
Comment 1:
The paper proposes a low-cost hyperspectral imaging system that combines GoPro cameras with LVSBPF, which has significant cost advantages. And successfully classified brown algae and red algae with highly overlapping morphological and spectral features through 1D-CNN, verifying the feasibility of the technology and providing new ideas for large-scale ecological monitoring through hyperspectral methods. Suggest highlighting the improvements made to the model and experimental results in the abstract section to demonstrate its innovation and feasibility.
Response 1:
Thank you for your suggestion. We agree that emphasizing the results will help demonstrate the feasibility of our study. We have updated the abstract to highlight the classification performance and the system’s capability to capture meaningful spectral information. As no modifications were made to the CNN architecture—since a standard 1D-CNN was used—we focused on reporting the classification outcomes. The revision has been made in the abstract section, lines 10–14.
"Our HSI system successfully captured spectral information from the target species, which exhibit considerable similarity in morphology and spectral profile, making them difficult to differentiate using traditional RGB imaging. Using a 1-dimensional convolutional neural network, we reached a high average classification precision, recall and F1-score of 99.9%, 89.5% and 94.4% respectively, demonstrating the effectiveness of our custom low-cost HSI setup."
Comment 2:
In terms of experimental design, the experiment was only conducted in a controlled water environment and the system performance was not validated in real ocean scenarios such as dynamic water flow, complex lighting, and suspended particle interference, which may overestimate the robustness of practical applications. And the dataset is divided into 90% for training and 10% for validating the trained model. What impact do different data partitioning methods have on the experimental results? Suggest the author to conduct appropriate analysis on the above two points.
Response 2.1:
Thank you for this constructive comment. We agree that evaluating the system under real ocean conditions is important for assessing its robustness. However, the focus of the current study was to explore first principles, i.e. whether a consumer-grade camera such as a GoPro can be adapted into a low-cost HSI system and to test its capability on a challenging classification task involving spectrally similar species. To maintain experimental control and isolate system performance, we limited the number of external variables. This has now been clarified in the discussion section (page 17, lines 490–492).
"Our aim has been to explore the integration of a consumer camera in a low-cost HSI system using controlled experimental settings and isolating system performance by limiting external variables."
Response 2.2
Thank you for the comment. Given the relatively small dataset, we selected a 90/10 split to maximize the training data, which is a common approach in similar contexts. We initially tested other partitioning ratios (80/20 and 70/30), but these resulted in lower performance—particularly in recall and F1-score generally being 13-16% lower. This clarification has been added to the Materials and Methods section, page 13, lines 341-344.
"The dataset was split into 90% for training and 10% for validation to support hyperparameter tuning. Alternative splits of 80/20 and 70/30 (training/validation) were also tested, but these configurations led to reduced model performance, particularly in recall and F1-score."
Comment 3:
From the experimental results, this article has provided a feasible low-cost solution for large-scale biological monitoring, and is innovative in the development and application of low-cost hyperspectral imaging technology. It is suggested to add a summary of the contributions made by each chapter of this article at the end of the introduction, in order to facilitate readers to have a general understanding of the system of this article during initial reading.
Response 3:
Agree. We have accordingly revised the manuscript to emphasize this suggestion, by adding a content summary at the end of the introduction. This is revised in the Introduction section (page 2, lines 83-88). "The remainder of this paper is structured as follows: Material and Methods Section details the optical setup and acquisition procedure and describes the data pre-processing pipeline used to construct hyperspectral cubes. The Results Section presents the spectral library and classification results obtained using the 1D-CNN model. Finally, the Discussion Section discusses spectral and classification results, the implications of our findings for scalable, low-cost ecological monitoring and outlines potential directions for future work."
Comment 4:
This article is a scientific and technological innovation paper. It is recommended that the author cite more papers from the past five years in terms of references to reflect the timeliness of this article. It is suggested that the author provide more detailed subheadings for each chapter of this article to reflect its innovative content.
Response 4:
We appreciate this helpful suggestion . We have included the following papers to highlight the timeliness of this paper.
Introduction section, page 2, line 72, end of sentence "...spectral bandpass filter (LVSBPF)
have been explored"
Shinatake, K.; Ishinabe, T.; Shibata, Y.; Fujikake, H. High-speed Tunable Multi-Bandpass Filter for Real-time Spectral Imaging
using Blue Phase Liquid Crystal Etalon. ITE Transactions on Media Technology and Applications 2020, 8, 202–209. https://doi.org/ 10.3169/mta.8.202.
Materials and Method section, page 3, line 105, end of sentence "...as is the case with push-broom
systems".
Zhang, X.; Li, S.; Xing, Z.; Hu, B.; Zheng, X. Automatic Registration of Remote Sensing High-Resolution Hyperspectral Images
Based on Global and Local Features. Remote Sensing 2025, 17. https://doi.org/10.3390/rs17061011.
Materials and Method section, page 12, line 311, end of sentence "...Based on previous studies on the classification of hyperspectral data"
Li, W.; Wang, Y.; Yu, Y.; Liu, J. Application of Attention-Enhanced 1D-CNN Algorithm in Hyperspectral Image and Spectral
Fusion Detection of Moisture Content in Orah Mandarin (Citrus reticulata Blanco). Information 2024, 15. https://doi.org/10.339 0/info15070408.
Discussion section, page 17, line 471, end of sentence "...The price of commercial HSI
cameras starts at 28.000 US dollars causing a barrier to in situ monitoring"
Pechlivani, E.M.; Papadimitriou, A.; Pemas, S.; Giakoumoglou, N.; Tzovaras, D. Low-Cost Hyperspectral Imaging Device for
Portable Remote Sensing. Instruments 2023, 7. https://doi.org/10.3390/instruments7040032.
Comments 5:
The subheading is too simple, such as software, Model, Suggest the author to rename it to reflect the research content of each section.
Response 5:
Agree, we have revised the subheadings to reflect the research content more clearly. at page 3, line 91, we have changed "2.1.1. Setup" to "2.1.1. Setup and Camera Settings"
At page 12, line 295, we have changed "2.4.1. Software" to "Software and Computer Specifications"
At page 12, line 310, we changed "2.4.3. Model" to "2.4.3. Model Description and Training Parameters"
At page 16, line 435, we changed "4.2 Model Classification" to "4.2 Model Performance and Classification"
Reviewer 2 Report
Comments and Suggestions for Authors
The paper presents an original method for improving standard RGB video measurements to hyperspectral measurements, which can be used not only to solve the problem of macroalgae monitoring presented by the authors, but also to solve any other problem where hyperspectral video measurements are required.
The quality of material presentation is high - radiometric peculiarities of measurements, peculiarities of illumination and background characteristics are taken into account. Possible problems that may arise during in-situ monitoring are considered.
There is some disadvantage of the work in that the number of analysed samples of macroalgae is not large enough, and it is desirable to take into account their possible seasonal variability and state, as well as to check how much it all depends on the characteristics of the surrounding sea waters, but this is not so important since the main result of the article is the developed device.
The paper fits the theme of the journal Sensors and can be published, but a small clarification is needed:
1) On what considerations was the CNN model setup chosen, e.g. number of layers, size of the convolutional kernel, etc.? Is it some kind of standard for similar problems or are the parameters chosen based on minimising selected statistical metrics to describe the quality of the model performance?
Author Response
Comments 1: On what considerations was the CNN model setup chosen, e.g. number of layers, size of the convolutional kernel, etc.? Is it some kind of standard for similar problems or are the parameters chosen based on minimising selected statistical metrics to describe the quality of the model performance?
Response: 1
Thank you for pointing this out. We agree that providing justification of the CNN architecture is important to improve clarity and reproducibility. We have therefore revised the manuscript (page 12, section 2.4.3: Model, line 324-326), to clarify:
"The architecture of our CNN largely follows that of [38], however, we removed the third convolutional and pooling layers, as initial experiments showed degraded performance with this additional layer".
The decision was made based on empirical results during model tuning. We also clarify this in the discussion section at page 16, lines 436-440, "In our study, we employed a 1D-CNN model with two convolutional layers. This gave
the best results compared to three convolutional layers as was mentioned in [ 38 ]. The third
layer resulted in poor prediction accuracy, potentially due to overfitting on the current
dataset. However, with larger or higher resolution 16-bit images, a third layer is likely
to enhance model performance."
Reviewer 3 Report
Comments and Suggestions for Authors
- The GoPro camera's automatic exposure function may change ISO and exposure parameters during measurement, causing calibration coefficients to become invalid. We suggest the authors specify whether the camera has been set to manual mode with all parameters locked, and verify these settings remain stable throughout the measurement process to ensure spectral measurement accuracy.
- LVSBPF's spectral scanning through physical movement may face issues with stepper motor precision, motion consistency, and mechanical vibrations, all affecting spectral data accuracy. We recommend the authors evaluate the stepper motor's precision and repeatability, analyze wavelength positioning errors at different scanning speeds, and verify the system's spectral reproducibility across multiple measurements.
- The paper needs to supplement the LVSBPF working principles, including how wavelength selection is achieved through film thickness gradients, its optical properties, and the mechanism of bandwidth variation with wavelength. We suggest adding filter structure diagrams, transmittance curve data, and discussing how non-ideal characteristics affect system performance.
- After LVSBPF spectral separation, light signal intensity significantly decreases, while the GoPro camera has limited dynamic range, potentially resulting in insufficient signal-to-noise ratio under low-light conditions. We suggest the authors analyze signal-to-noise ratios at different wavelengths, evaluate camera performance in low-light conditions, and consider optimizing illumination or integration time to ensure reliable data across the entire spectral range.
Author Response
Comment 1:
The GoPro camera's automatic exposure function may change ISO and exposure parameters during measurement, causing calibration coefficients to become invalid. We suggest the authors specify whether the camera has been set to manual mode with all parameters locked, and verify these settings remain stable throughout the measurement process to ensure spectral measurement accuracy.
Response 1:
We appreciate the helpful suggestion. We confirm that the GoPro camera was set to manual mode, with ISO and shutter speed parameters locked throughout the measurements. Additionally, we verified the stability of these settings by reviewing the spectral data after each scan to ensure consistency and uniformity across all measurements. We have clarified this in at the Material and Method section, page 4, lines 146-148, "The GoPro camera was operated in Manual mode, recording 2704×1520 pixels images with fixed parameters; 240 Hz frame rate, 1/240 s exposure time and ISO=100. The parameters
where locked to ensure consistency across spectral measurements"
Comment 2:
LVSBPF's spectral scanning through physical movement may face issues with stepper motor precision, motion consistency, and mechanical vibrations, all affecting spectral data accuracy. We recommend the authors evaluate the stepper motor's precision and repeatability, analyze wavelength positioning errors at different scanning speeds, and verify the system's spectral reproducibility across multiple measurements.
Response 2:
Thank you for this relevant and thoughtful comment. We fully acknowledge the potential impact of stepper motor precision, motion consistency, and mechanical vibrations on spectral accuracy. However, varying the scanning speed would alter both the reconstruction process and the spectral resolution, which would no longer align with the acquisition parameters consistently applied throughout this study. The main objective of this work is to demonstrate the feasibility of integrating an LVSBPF with a GoPro camera for capturing spectral data suitable for classification tasks. While a full mechanical evaluation is outside the scope of this study, we conducted preliminary tests to assess repeatability. Spectral measurements from the same spatial point across three full filter sweeps showed consistent spectral profiles, suggesting stable performance of the LVSBPF under the current setup. We have now clarified this point in the revised manuscript on page 3 , line 108-111: “To evaluate the stability of the stepper motor mechanism with the LVSBPF, repeated measurements of a fixed spatial point across three full spectral sweeps were compared, showing minimal variation in the resulting spectra, thus indicating reliable filter performance under the defined scanning parameters.”
Comment 3:
The paper needs to supplement the LVSBPF working principles, including how wavelength selection is achieved through film thickness gradients, its optical properties, and the mechanism of bandwidth variation with wavelength. We suggest adding filter structure diagrams, transmittance curve data, and discussing how non-ideal characteristics affect system performance.
Response 3:
Thank you for the suggestion. The LVSBPF used in this study is a commercially available filter (Model XXX, Delta Optical Thin Film A/S), not developed in-house and specific information about the filter can be found at their webside. We have therefore clarified this in the revised manuscript page 3, line 93, "...(For details about the filter, please find the model, 400-700nm, 3-9nm bandwidth, at Delta Optical Thin Film A/S)".
Filter link: https://deltaopticalthinfilm.com/products/continuously-variable-filters/continuously-variable-bandpass-filters/ - the model is the 400-700nm with 3-9nm bandwidth. The spectral response is below:
Comment 4:
After LVSBPF spectral separation, light signal intensity significantly decreases, while the GoPro camera has limited dynamic range, potentially resulting in insufficient signal-to-noise ratio under low-light conditions. We suggest the authors analyze signal-to-noise ratios at different wavelengths, evaluate camera performance in low-light conditions, and consider optimizing illumination or integration time to ensure reliable data across the entire spectral range.
Response 4:
Thank you for this valuable comment. We acknowledge the limitations of the GoPro's dynamic range, particularly under reduced light conditions after spectral separation. As the camera operates in manual mode, exposure time remains fixed throughout the acquisition of the datacube to ensure consistency and comparability across all wavelength bands. Adjusting exposure dynamically would require switching to automatic mode, which unfortunately removes user control and transparency over the settings - compromising data reliability.
We agree that signal-to-noise ratio optimization is essential, especially for field deployments. This is an important consideration for future work involving real-world scenarios, where improvements in illumination and sensor control will be necessary.